# Management of childbearing with hypermobile Ehlers-Danlos syndrome and hypermobility spectrum disorders: A scoping review and expert co-creation of evidence-based clinical guidelines

Sally Pezaro [1,2]*, Isabelle Brock[3], Maggie Buckley[4], Sarahann Callaway[5], Serwet Demirdas[6], Alan Hakim[7], Cheryl Harris[8], Carole High Gross[9], Megan Karanfil[10], Isabelle Le Ray[11], Laura McGillis[12], Bonnie Nasar[13], Melissa Russo[14], Lorna Ryan[15], Natalie Blagowidow[16]

1 Research Centre for Healthcare and Communities, Coventry University, Coventry, United Kingdom, 2 The University of Notre Dame, Notre Dame, Australia, 3 Department of Connective Tissue, Nova Combian Research Institute, New York, New York, United States of America, 4 The Ehlers Danlos Society's International Consortium, New York, New York, United States of America, 5 Main Line Health- Bryn Mawr Rehab, King of Prussia, Pennsylvania, United States of America, 6 Department of Clinical Genetics, Erasmus Medical Centre, Rotterdam, The Netherlands, 7 The Ehlers-Danlos Society, The Ehlers-Danlos Society – Europe, London, United Kingdom, 8 Harris Whole Health, Fairfax, Virginia, United States of America, 9 Lehigh Valley Health Network, Palmer, Pennsylvania, United States of America, 10 The International Consortium on the Ehlers-Danlos syndromes and Hypermobility Spectrum Disorders, The Herds Nerd, Baltimore, Maryland, United States of America, 11 Integrative Systemic Medicine Center, Boulogne-Billancourt and Strasbourg University Hospital, Strasbourg, France, 12 GoodHope EDS Program, Toronto General Hospital, Toronto, Ontario, Canada, 13 Registered Dietitian Nutritionist, Ridgewood, New Jersey, United States of America, 14 Women and Infants Hospital, An Affiliate of Warren Alpert Medical School at Brown University in Providence, Providence, Rhode Island, United States of America, 15 Lorna Ryan Health, London, United Kingdom, 16 Harvey Institute for Human Genetics, Greater Baltimore Medical Center, Baltimore, Maryland, United States of America

* sally.pezaro@coventry.ac.uk

## Abstract

### Objective

To co-create expert guidelines for the management of pregnancy, birth, and postpartum recovery in the context of hypermobile Ehlers-Danlos syndrome (hEDS) and hypermobility spectrum disorders (HSD).

### Design

Scoping Review and Expert Co-creation.

### Setting

United Kingdom, United States of America, Canada, France, Sweden, Luxembourg, Germany, Italy, and the Netherlands.

**Data Availability Statement:** All relevant data are within the paper and its Supporting information files.

**Funding:** The author(s) received no specific funding for this work.

**Competing interests:** SP reports receiving honorariums from the Ehlers-Danlos Society. AH reports receiving honorariums from the Ehlers-Danlos Society. NB reports receiving honorariums from the Ehlers-Danlos Society. Other authors report no conflict of interest.

## Sample

Co-creators (n = 15) included expertise from patients and clinicians from the International Consortium on the Ehlers-Danlos syndromes and Hypermobility Spectrum Disorders, facilitated by the Ehlers-Danlos Society.

## Methods

A scoping review using Embase, Medline, the Cochrane Central Register of Controlled Trials and CINHAL was conducted from May 2022 to September 2023. Articles were included if they reported primary research findings in relation to childbearing with hEDS/HSD, including case reports. No language limitations were placed on our search, and our team had the ability to translate and screen articles retrieved in English, French, Spanish, Italian, Russian, Swedish, Norwegian, Dutch, Danish, German, and Portuguese. The Mixed Methods Appraisal Tool was used to assess bias and quality appraise articles selected. The co-creation of guidelines was based on descriptive evidence synthesis along with practical and clinical experience supported by patient and public involvement activities.

## Results

Primary research studies (n = 14) and case studies (n = 21) including a total of 1,260,317 participants informed the co-creation of guidelines in four overarching categories: 1) Preconceptual: conception and screening, 2) Antenatal: risk assessment, management of miscarriage and termination of pregnancy, gastrointestinal issues and mobility, 3) Intrapartum: risk assessment, birth choices (mode of birth and intended place of birth), mobility in labor and anesthesia, and 4) Postpartum: wound healing, pelvic health, care of the newborn and infant feeding. Guidelines were also included in relation to pain management, mental health, nutrition and the common co-morbidities of postural orthostatic tachycardia syndrome, other forms of dysautonomia, and mast cell diseases.

## Conclusions

There is limited high quality evidence available. Individualized strategies are proposed for the management of childbearing people with hEDS/HSD throughout pregnancy, birth, and the postpartum period. A multidisciplinary approach is advised to address frequently seen issues in this population such as tissue fragility, joint hypermobility, and pain, as well as common comorbidities, including dysautonomia and mast cell diseases.

## Introduction

The Ehlers-Danlos syndromes (EDS) are a group of underdiagnosed, heritable connective tissue disorders characterized by generalized joint hypermobility (GJH), skin hyperextensibility and tissue fragility. Thirteen types of EDS were identified by the International Consortium on the Ehlers-Danlos Syndromes in 2017 [1], and a fourteenth subtype has since been identified [2]. Hypermobile EDS (hEDS) is the most common type of EDS, mainly symptomatic in people assigned female at birth [3,4]. Updated diagnostic criteria for hEDS released in 2017 narrowly define hEDS based on the presence of GJH, multiple features of underlying connective tissue weakness, with or without confirmed family history of the same diagnosis [1]. Those

**Table 1. Signs and symptoms of EDS (all subtypes) in females with a frequency ≥70% (34 out of 79 signs).**

| Signs and symptoms | Frequency |
|---|---|
| Fatigue, hypermobility, arthralgia, delicate/ transparent skin, sleeping problems | 90–97% |
| Meno/metrorrhagia, myalgia/cramps, dysfunctional proprioception, skin bleeding, migraines or headaches, visual fatigue, joint dislocations, plantar contractions, dyspnoea, temperature dysregulation | 80–89% |
| Genital bleeding, genital pain, sprains or pseudo-sprains, pseudo-Raynaud phenomenon, difficult wound healing, cutaneous hyperesthesia, abdominal pain, gastroesophageal reflux, bloating, or distension caused by gas, hyperhidrosis, attention deficit, decreased working memory capacity, upper respiratory infections, hypersomnia, vertigo | 70–79% |

presenting with GJH, musculoskeletal complications and pain with or without associated comorbidities, who do not fulfill criteria of hEDS may fall under the diagnostic category of hypermobility spectrum disorders (HSD), for which the same management approaches will apply [5,6]. Thus hEDS/HSD are referred to together throughout this article, though we recognize that more will be diagnosed with HSD than hEDS. hEDS/HSD is likely underdiagnosed overall but recent estimations suggest that the combined prevalence of hEDS and HSD in the order of 1 in 600 to 1 in 900 [4], with many pregnancies affected [7]. Separate prevalence estimates for hEDS and HSD are not currently available. Patients with hEDS/HSD present with multisystem signs, symptoms, and comorbidities the most common of which are outlined in Table 1, adapted from Hamonet and colleagues with permission [8].

Regarding frequencies presented in Table 1, the control group had symptoms at <10% frequency, with the following exceptions: sleeping problems (26%), upper respiratory infections (23%), bloating or distension caused by gas (19%), gastroesophageal reflux (13%), and meno/metrorrhagia (11%).

Prevalence of obstetric, pelvic and reproductive system issues reported by those childbearing with hEDS/HSD are significantly greater than in the general childbearing population [9,10]. Symptoms can be debilitating; they may either be exacerbated or improve during pregnancy and birth, when hormonal levels are elevated [7,11–13]. Still, there is scarce literature related to perinatal care in the context of hEDS/HSD, leading to potential misconceptions and a lack of knowledge in some health care professionals [7,14]. Discussions with patient groups and members of the International Consortium on the Ehlers-Danlos syndromes and Hypermobility Spectrum Disorders established the importance of and need to have international expert guidelines drawing from existing evidence and expert opinion to provide guidance to care providers and persons with these conditions in childbearing. Moreover, evidence-based guidelines have been most commonly requested by both perinatal professionals and those childbearing with hEDS/HSD in a large international survey investigating perinatal staff's knowledge and confidence in supporting people with hEDS/HSD, and people with hEDS/HSD's experiences of perinatal care [14].

Considering the above, the overarching aim of this research was to co-create international evidence-based expert guidelines on the management of pregnancy, birth, and post-natal recovery in the context of hEDS/HSD. A scoping review was conducted to inform the co-creation of these guidelines and answer the following question: What is known from the literature about pregnancy, birth, and post-natal recovery in people with hEDS/HSD?

Scoping reviews differ from systematic reviews and lend themselves to scoping a body of literature while utilising the experience and expertise of their authors [15]. They are also complementary to co-creating guidelines and guidelines such as these [16] and are recommended to addresses several questions from a diverse body of literature pertaining to a broad topic. As such, a scoping review was considered the most rigorous approach to evaluate the evidence

and inform these guidelines, rather than a systematic review, which is methodologically suited to address only one question. Our scoping review followed a preferred framework [17], consisting of the following steps: (1) identifying the research question(s), (2) identifying relevant studies, (3) study selection, (4) charting the data, (5) collating, summarising, and reporting the results, and (6) optional consultation exercises. We also included patient representatives and ensured stakeholder engagement throughout to enhance the rigor of our scoping approach [18]. This also met our objective with regards to using the findings of a scoping review to inform the co-creation of evidence-based guidelines in this context.

## Methods

This research was undertaken using evidence-based approaches [19]. Alongside a scoping review, the three Co's framework of 'Co-define' (examining problems and positive aspects), 'Co-design' (prioritising problems and designing solutions), 'Co-refine' (co-produce and refine together) was used to co-create the guidance presented [20]. Such co-creation is underpinned by participatory action research [21–23] and design thinking [24,25]. In person consensus meetings along with Patient and Public Involvement (PPI) activities involving wider patient and public groups and convened by the Ehlers-Danlos Society were also undertaken in finalizing the guidelines presented.

Co-creator recruitment began after ethical approval was granted via the lead author's University in March 2022 (Project P135062). Our sampling strategy was purposive. Members of the International Consortium on the Ehlers-Danlos syndromes and Hypermobility Spectrum Disorders (www.ehlers-danlos.com/international-consortium), including those with lived experience of both hEDS/HSD and childbearing, along with clinicians and academics were sent participant information via email and were invited to participate.

Once informed consent was secured, each co-creator was given access to the project's co-creation space online and invited to visit the 'co-creating welfare' project website (http://ccw. southdenmark.eu/) to become familiar with the principles of co-creation. Those who gave their consent to participate were invited to engage in a series of online co-creation workshops hosted by the Ehlers-Danlos Society and led by the principal author.

During the first co-creation workshop, activities focused on 'co-defining' what the issues, problems, and positive aspects of perinatal care in this context are. Thereafter, co-creators were invited to prioritise the problems identified, find solutions, and 'co-design' expert guidance together in real time. These were then co-produced and refined together. Both during and outside of bi-weekly 'co-refining' workshops, co-creators were invited to 'co-refine' guidelines via an iterative succession of discussions and annotated co-refinements.

Co-creation began in March 2022 and concluded in November 2023 following a face-to-face consensus meeting held in Arizona (August 2022), and public and patient involvement activities held in Rome (September 2022) hosted by the Ehlers-Danlos Society. During 2023, four further co-refining workshops were hosted to discuss residual deviating views and ideas until unanimity was reached. In line with our evidence-based approach [19], co-creation activities were supported by our scoping review, which was last updated in September 2023. Expert co-creators consulted the findings of each article included and used these, along with their practical knowledge and/or experiences to support and inform the co-created guidelines.

The search strategy for our scoping review is outlined in Table 2 and was formulated in partnership with librarians at the Erasmus MC Medical Library in Rotterdam in the Netherlands, inclusive of previous nomenclature. The PRISMA extension for scoping reviews (PRISMA-ScR) was used to guide reporting (See checklist in S1 File).

**Table 2. Search strategy.**

**Two search elements:**

1. hypermobile ehlers danlos (type 3; hypermobility spectrum disorders, ht)

2. pregnancy

Embase (n = 171)

(('Ehlers Danlos syndrome'/de AND ('hypermobility'/de OR 'joint mobility'/de OR 'hyperlaxity'/de)) OR 'ehlers danlos syndrome hypermobility type'/de OR 'hypermobility syndrome'/de OR 'hypermobility spectrum disorder'/de OR 'hypermobile ehlers danlos syndrome'/de OR (((type-3 OR type3 OR type-III OR typeIII OR hypermobil* OR ht OR h-t OR joint-mobil* OR joint-flexib* OR range-of-motion* OR hyperlaxit*) NEAR/9 (ehlers*) NEAR/9 (danlos*)) OR hEDS OR h-EDS OR EDS3 OR EDS-3 OR EDS-III OR EDSIII OR htEDS OR h-tEDS OR ((type-3 OR type3 OR type-III OR typeIII OR hypermobil* OR ht OR h-t OR joint-mobil* OR joint-flexib* OR range-of-motion* OR hyperlaxit*) NEAR/9 (EDS OR ehlers OR danlos)) OR ((hypermobilit*) NEAR/3 (disorder* OR syndrom*)):ab,ti,kw) AND ('pregnancy'/de OR 'pregnant woman'/de OR 'prenatal care'/de OR 'obstetric delivery'/exp OR 'birth'/exp OR 'pregnancy complication'/de OR 'gynecology'/de OR 'uterus'/exp OR 'labor'/de OR 'prepregnancy care'/de OR 'breast feeding'/de OR 'infant feeding'/de OR 'cesarean section'/de OR 'expectant mother'/de OR (pregnan* OR gestation* OR antenatal* OR intrapartum* OR postnatal OR postpartum OR ((post OR ante OR intra) NEAR/3 (natal* OR partum* OR partum*)) OR delivery OR deliveries OR parturition* OR birth* OR childbirth* OR gynecolog* OR gynaecolog* OR obstetric* OR uter* OR labor OR labour OR pre-concept* OR preconcept* OR prepregnan* OR ((breast* OR infant*) NEAR/3 (feed*)) OR cesarean* OR caesarean* OR expectant-mother*):ab,ti,kw,jt)

Medline (n = 110)

((Ehlers-Danlos Syndrome/ AND Range of Motion, Articular/) OR Ehlers-Danlos syndrome type 3.rs. OR (((type-3 OR type3 OR type-III OR typeIII OR hypermobil* OR ht OR h-t OR joint-mobil* OR joint-flexib* OR range-of-motion* OR hyperlaxit*) ADJ9 (ehlers*) ADJ9 (danlos*)) OR hEDS OR h-EDS OR EDS3 OR EDS-3 OR EDS-III OR EDSIII OR htEDS OR h-tEDS OR ((type-3 OR type3 OR type-III OR typeIII OR hypermobil* OR ht OR h-t OR joint-mobil* OR joint-flexib* OR range-of-motion* OR hyperlaxit*) ADJ9 (EDS OR ehlers OR danlos)) OR ((hypermobilit*) ADJ3 (disorder* OR syndrom*))).ab,ti,kf.) AND (exp Pregnancy/ OR Pregnant Women/ OR Prenatal Care/ OR exp Delivery, Obstetric/ OR exp Parturition/ OR exp Pregnancy Complications/ OR Gynecology/ OR exp Uterus/ OR exp Labor, Obstetric/ OR exp Breast Feeding/ OR (pregnan* OR gestation* OR antenatal* OR intrapartum* OR postnatal OR postpartum OR ((post OR ante OR intra) ADJ3 (natal* OR partum* OR partum*)) OR delivery OR deliveries OR parturition* OR birth* OR childbirth* OR gynecolog* OR gynaecolog* OR obstetric* OR uter* OR labor OR labour OR pre-concept* OR preconcept* OR prepregnan* OR ((breast* OR infant*) ADJ3 (feed*)) OR cesarean* OR caesarean* OR expectant-mother*).ab,ti,kf,jw.)

Cochrane (n = 5)

(((((type NEXT 3 OR type3 OR type NEXT III OR typeIII OR hypermobil* OR ht OR h NEXT t OR joint NEXT mobil* OR joint NEXT flexib* OR range NEXT of NEXT motion* OR hyperlaxit*) NEAR/9 (ehlers*) NEAR/9 (danlos*)) OR hEDS OR h NEXT EDS OR EDS3 OR EDS NEXT 3 OR EDS NEXT III OR EDSIII OR htEDS OR h NEXT tEDS OR ((type NEXT 3 OR type3 OR type NEXT III OR typeIII OR hypermobil* OR ht OR h NEXT t OR joint NEXT mobil* OR joint NEXT flexib* OR range NEXT of NEXT motion* OR hyperlaxit*) NEAR/9 (EDS OR ehlers OR danlos)) OR ((hypermobilit*) NEAR/3 (disorder* OR syndrom*))):ab,ti,kw) AND ((pregnan* OR gestation* OR antenatal* OR intrapartum* OR postnatal OR postpartum OR ((post OR ante OR intra) NEAR/3 (natal* OR partum* OR partum*)) OR delivery OR deliveries OR parturition* OR birth* OR childbirth* OR gynecolog* OR gynaecolog* OR obstetric* OR uter* OR labor OR labour OR pre NEXT concept* OR preconcept* OR prepregnan* OR ((breast* OR infant*) NEAR/3 (feed*)) OR cesarean* OR caesarean* OR expectant NEXT mother*):ab,ti,kw)

Cinahl (n = 53)

((MH Ehlers-Danlos Syndrome AND (MH Joint Instability OR MH Range of Motion)) OR TI(((type-3 OR type3 OR type-III OR typeIII OR hypermobil* OR ht OR h-t OR joint-mobil* OR joint-flexib* OR range-of-motion* OR hyperlaxit*) N9 (ehlers*) N9 (danlos*)) OR hEDS OR h-EDS OR EDS3 OR EDS-3 OR EDS-III OR EDSIII OR htEDS OR h-tEDS OR ((type-3 OR type3 OR type-III OR typeIII OR hypermobil* OR ht OR h-t OR joint-mobil* OR joint-flexib* OR range-of-motion* OR hyperlaxit*) N9 (EDS OR ehlers OR danlos))) OR AB(((type-3 OR type3 OR type-III OR typeIII OR hypermobil* OR ht OR h-t OR joint-mobil* OR joint-flexib* OR range-of-motion* OR hyperlaxit*) N9 (ehlers*) N9 (danlos*)) OR hEDS OR h-EDS OR EDS3 OR EDS-3 OR EDS-III OR EDSIII OR htEDS OR h-tEDS OR ((type-3 OR type3 OR type-III OR typeIII OR hypermobil* OR ht OR h-t OR joint-mobil* OR joint-flexib* OR range-of-motion* OR hyperlaxit*) N9 (EDS OR ehlers OR danlos)))) AND (MH Pregnancy+ OR MH Expectant Mothers OR MH Prenatal Care OR MH Delivery, Obstetric+ OR MH Pregnancy Complications+ OR Gynecology+ OR MH Uterus+ OR MH Labor+ OR MH Breast Feeding+ OR MH Obstetrics OR TI(pregnan* OR gestation* OR antenatal* OR intrapartum* OR postnatal OR postpartum OR ((post OR ante OR intra) N2 (natal* OR partum* OR partum*)) OR delivery OR deliveries OR parturition* OR birth* OR childbirth* OR gynecolog* OR gynaecolog* OR obstetric* OR uter* OR labor OR labour OR pre-concept* OR preconcept* OR prepregnan* OR ((breast* OR infant*) N2 (feed*)) OR cesarean* OR caesarean* OR expectant-mother*) OR AB(pregnan* OR gestation* OR antenatal* OR intrapartum* OR postnatal OR postpartum OR ((post OR ante OR intra) N2 (natal* OR partum* OR partum*)) OR delivery OR deliveries OR parturition* OR birth* OR childbirth* OR gynecolog* OR gynaecolog* OR obstetric* OR uter* OR labor OR labour OR pre-concept* OR preconcept* OR prepregnan* OR ((breast* OR infant*) N2 (feed*)) OR cesarean* OR caesarean* OR expectant-mother*) OR JT(pregnan* OR gestation* OR antenatal* OR intrapartum* OR postnatal OR postpartum OR ((post OR ante OR intra) N2 (natal* OR partum* OR partum*)) OR delivery OR deliveries OR parturition* OR birth* OR childbirth* OR gynecolog* OR gynaecolog* OR obstetric* OR uter* OR labor OR labour OR pre-concept* OR preconcept* OR prepregnan* OR ((breast* OR infant*) N2 (feed*)) OR cesarean* OR caesarean* OR expectant-mother*))

Screening of the articles was led by two members of the team (SP and ILR) who engaged all co-creators in decision making during bi-weekly meetings. Final articles were included to inform the guidelines if they reported primary research findings in relation to childbearing with hEDS/HSD. Case reports were also included to inform the guidelines presented. No limits were placed with regards to the language of articles to be screened, and our team had the ability to translate and screen articles retrieved in the following languages: English, French, Spanish, Italian, Russian, Swedish, Norwegian, Dutch, Danish, German, and Portuguese. Articles were excluded if they did not relate to childbearing in the context of hEDS/HSD. Alongside the

inclusion of newly published peer reviewed articles shared via professional networks, the reference lists of identified articles and literature reviews were also screened for additional relevant citations. Reviews of the literature were excluded if they did not also feature a clinical case study. Authors of all selected articles were invited to offer any further available evidence for inclusion. To further enhance the rigor of our approach, primary research articles (excluding case studies) which met inclusion criteria were then quality appraised using the Mixed Methods Appraisal Tool (MMAT), which enables researchers to simultaneously evaluate the validity and reliability of both quantitative and qualitative empirical studies [26]. Quality scores range from * if one criterion is met to ***** if all five criteria are met [27]. Final appraisal scores were proposed in consultation by 3 members of the team (ILR, SD and SP), and then agreed by the wider co-creation team.

Co-creators considered the identified evidence and contextualized it alongside the co-production of guidelines while engaging the principles of the WHO-INTEGRATE evidence to decision framework [28]. In line with best practice and where evidence was lacking [19], co-creators formulated guidelines based on practical and clinical experience, and with input from those with lived experiences. Once guidelines were finalized, external members of the International Consortium with relevant subject expertise distinct from those within the co-creation team were invited to assess the final guidelines for clarity and relevance.

This research occurred in a context where minoritized Black and other ethnic communities, the Lesbian, Gay, Bisexual, Transgender, Queer and/or Questioning, Intersex, Asexual, Two-Spirit (LGTBQIA2S+) communities and those with low socioeconomic status face challenges and experience discrimination in healthcare every day [29]. Consequently, gender-inclusive and person-first language was used throughout reporting. All co-creators were unanimous in approving the finalized guidelines reported here.

## Patient and public involvement activities

Patient and stakeholder engagement in guideline development is internationally advocated [30]. We report the Patient and Public Involvement (PPI) activities conducted in line with the Guidance for Reporting Involvement of Patients and the Public (GRIPP2) short form checklist [31]. To reduce the potential for bias, prior relationships with those engaging in PPI activities and the co-creation team were minimal, and activities remained congruent with best practice [30].

**Aim.** The aim of PPI in this study was to involve members of the public and those with lived experience of hEDS/HSD and childbearing in the co-creation of international expert guidelines for the management of pregnancy, birth, and post-natal recovery in the context of hEDS/HSD. We were also keen to co-author and disseminate this publication with those who have had experience of childbearing with hEDS/HSD.

**Methods.** Patient representatives were involved in this project from its conception, both as co-creators, authors, and consultants with lived experience of hEDS/HSD and/or childbirth. All patients and public attending conferences hosted by the Ehlers-Danlos Society in both Arizona (August 2022) and Rome (September 2022) were invited to comment upon and shape guidelines in partnership with the co-creation team in person. Further comments were also solicited via the Ehlers-Danlos Society's Global Affiliation Program newsletter, and via a separate EDS affiliate focus group facilitated online by MB including 12 participants representing USA, Canada, Sweden, Luxembourg, and Germany. All comments resulting from these activities were collated together and shared with the co-creation team. They were then examined collectively to identify common topics, ideas, and patterns of meaning which were then grouped into themes.

**Results.** Themes related to 'key topics' for inclusion in the guidelines and related to the management of miscarriage and termination of pregnancy, comorbidities, causal relationships, anesthesia, incontinence, and risk assessments along with the management of symptoms such as enhanced joint laxity and pain. Those who engaged in PPI requested guidelines from the co-creation team with regards to the preconceptual, antenatal, intrapartum, and postpartum periods.

**Discussion and conclusions.** PPI influenced this study substantially overall. A distinctly positive aspect of engaging in PPI from the start meant that we could stay focussed on what mattered most and communicate to professionals using the voices of those with lived experience. Nevertheless, it was challenging to balance the knowledge, voices, and concerns of experts with lived experience and practicing clinicians.

**Reflections.** Due to previous negative experiences with clinicians in healthcare, some engaged with PPI activities were doubtful as to the ability of clinicians to use their initiative or provide basic, competent, and compassionate care, and thus requested that guidelines included basic principles of healthcare practice. This was frustrating to clinical professionals, who expressed fears that clinicians would disengage from guidelines which repeated their basic training. The rebuilding of trust between clinicians and those with lived experience in this area may remove this as a barrier to future PPI activities.

## Results

A total of 15 co-creators joined in meeting the aim of this research from the United Kingdom, United States of America, Canada, France, and the Netherlands. Co-creators represented a variety of professions including midwifery, obstetrics, maternal fetal medicine, rheumatology, registered dietitians, and nutritionists, physical therapy, clinical geneticists, nursing, clinical research, and pain management. Co-creators also included patient advocates with lived experience of both hEDS/HSD and childbearing. All co-creators joined in authoring the final report and guidelines, and thus are listed as co-authors to this article.

Fig 1 outlines how the final articles (n = 35) resulting from our literature searches were identified for inclusion following the removal of duplicates.

The final articles which met our inclusion criteria consisted of primary research studies (n = 14) and case studies (n = 21). They included a total of 1,260,317 participants. Our collation and summarising of results are reported in Table 3, which presents article details, findings, and quality appraisal scores.

Final expert guidelines were broadly grouped into the following categories presented in Table 4: Preconceptual (conception and screening), Antenatal (risk assessment, management of miscarriage and termination of pregnancy and mobility), Intrapartum (risk assessment, birth choices, mode of birth/intended place of birth, mobility in labour and anaesthesia), and Postpartum (wound healing, pelvic health, care of the newborn and infant feeding). Our supplement also outlines guidance in relation to physical therapy (see table in S2 File). These are intended to act as guides rather than policy in the understanding that professionals take an individualized approach to care.

## Discussion

This article presents an international consensus of expert guidelines in the management of childbearing with hEDS/HSD brought about using systematic and co-creation approaches along with a variety of PPI activities. While we have built upon previous care considerations [11–13], these guidelines represent the first of their kind from an international multidisciplinary collaboration, including patient perspectives. In our experts' view, many of the guidelines

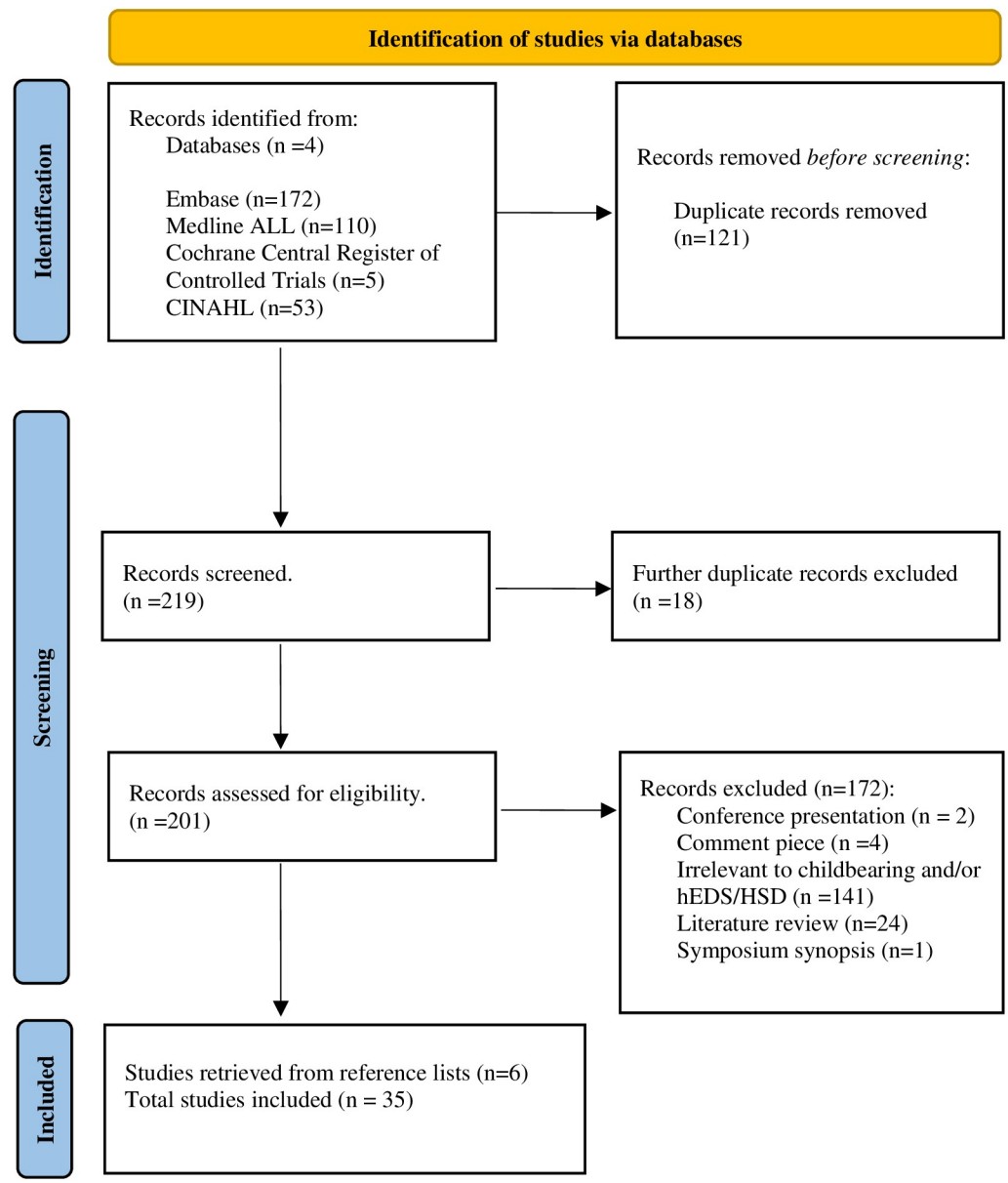

**Fig 1. PRISMA Flow diagram of database search outcomes.**

made can be applicable in other subtypes of EDS (except vascular EDS), though each type should be considered individually via future research. Practitioners may also consider these expert guidelines for those with generalized HSD, as underlying care principles will be largely the same. Moreover, as JHS is indistinguishable from hEDS/HSD and considered likely allelic, this guideline should also apply to those diagnostically labelled as having JHS. We also offer strategies in relation to managing the comorbidities associated with hEDS/HSD, although we recognize that further research is required in these areas, and more detailed expert consensus' remains warranted [73,74,123].

Guidelines were co-created into the following categories: Preconceptual (conception and screening), Antenatal (risk assessment, management of miscarriage and termination of

**Table 3. Selected article details and appraisal scores.**

| Article | Country of Origin | Study type and participants | Collation and summary of results | MMAT Score (* if one criterion is met; ***** if all five criterions are met) |
|---|---|---|---|---|
| **Empirical Articles** | | | | |
| Kanjwal et al. (2010) [32] | United States of America | Retrospective cohort study of patients with POTS with features of Joint Hypermobility Syndrome (JHS) (by Beighton criterion) (n = 26). | • Migraine was a common comorbidity 73 vs 29% p = 0.001.<br>• In two patients POTS was precipitated by pregnancy, and in three by surgery, urinary tract infection and a viral syndrome respectively.<br>• The common clinical features were fatigue (58%), orthostatic palpitations (54%), presyncope (58%), and syncope (62%).<br>• Patients with POTS and JHS appear to become symptomatic at an earlier age compared to POTS patients without JHS. In addition, patients with JHS had a greater incidence of migraine and syncope than their non JHS counterparts. | ***** |
| Pearce et al. (2023) [10] | UK, USA Australia, Canada, New Zealand, and Ireland | A large online international survey was completed by participants with experience in childbearing and a diagnosis of hEDS/HSD (n = 947, total pregnancies = 1338). | • Of babies born after 24 weeks ($N = 1230$), 190 (15.45%) were pre-term (occurring before 37 weeks) including 7 stillbirths.<br>• Of the 183 pre-term live births, 149 were singleton pregnancies (80 vaginal births; 69 caesarean births) and 34 were twins (7 vaginal births; 27 caesarean births)<br>• There were 44.80% ($N = 551$; vaginal = 358; caesarean = 193) births at term before the due date (between $\geq$37 weeks and <40 weeks) and 39.76% ($N = 489$; vaginal = 402; caesarean = 87) births after the due date ($\geq$40 weeks)<br>• Incidences were higher in people with hEDS/HSD than typically found in the general population for: pre-eclampsia, eclampsia, pre-term rupture of membranes, pre-term birth, antepartum hemorrhage, postpartum hemorrhage, hyperemesis gravidarum, shoulder dystocia, caesarean wound infection, postpartum psychosis, post-traumatic stress disorder, precipitate labor and being born before arrival at place of birth | ***** |

*(Continued)*

**Table 3.** (Continued)

| Article | Country of Origin | Study type and participants | Collation and summary of results | MMAT Score (* if one criterion is met; ***** if all five criterions are met) |
|---------|-------------------|-----------------------------|----------------------------------|------------------------------------------------------------------------------|
| Wright et al. (2023) [33] | United States of America | Retrospective cohort study of people with EDS (all types) birthing between the years 2016 and 2020 (n = 7378) | • Prevalence of EDS in pregnancy was 4.1 per 10,000 births, with the trend increasing over the 5-year study period from 2.7 to 5.2 per 10,000 birth related hospitalizations.<br>• Those with EDS were more likely to live in a top median ZIP code quartile.<br>• Pregnancies in those with EDS were associated with prematurity (aOR 1.43, 95% CI 1.23, 1.65), severe morbidity in the birthing parent (aOR 1.79, 95% CI 1.34, 2.38) and cervical insufficiency (aOR 2.17, 95% CI 1.48, 3.18).<br>• Those with EDS were more likely to give birth via caesarean section, (aOR 1.29, 95% CI 1.19, 1.40), have a postpartum hemorrhage, (aOR 1.31, 95% CI 1.10, 1.57), in both unadjusted (OR 1.76, 95% CI 1.37, 2.25) and adjusted models (aOR 1.78, 95% CI 1.39, 2.29).<br>• Of the 6117 with EDS evaluated for postpartum readmission, 3% were readmitted, 60% of these within 10 days of birth related hospitalization, vs 1.7% non-EDS births, (aOR 1.78, 95% CI 1.39, 2.29). | ***** |
| Castori, et al. (2012) [34] | Italy | A cohort study of 42 childbearing people with Joint Hypermobility Syndrome having had one or more pregnancy. Total conceptions (n = 93). | • Spontaneous abortions (16.1%)<br>• Voluntary interruptions (6.5%)<br>• Preterm births (10.7%)<br>• births at >37 weeks gestation (66.7%)<br>• Non-operative vaginal births (72.2%)<br>• Forceps/vacuum use (5.5%) caesarean births (22.3%)<br>• Abnormal scar formation after caesarean or episiotomy (46.1%)<br>• Hemorrhage (19.4%)<br>• Pelvic prolapses (15.3%)<br>• Deep venous thrombosis (4.2%)<br>• Coccyx dislocation (1.4%)<br>• Prolapses were the most clinically relevant complication and associated with episiotomy.<br>• Overall outcomes were good with no stillbirth and fetal/neonatal hypoxic/ischemic events.<br>• Local/total anesthesia was successfully performed in 17 pregnancies without any problem. | ***** |
| Hugon-Rodin et al. (2016) [35] | France | Cohort study including 386 consecutive people diagnosed with hEDS/HSD. | • Dyspareunia (43%).<br>• Birth via caesarean (14.6%) Premature births (6.2%)<br>• Multiple spontaneous abortion (13%)<br>• Spontaneous abortion (28%) | ***** |

*(Continued)*

**Table 3.** (*Continued*)

| Article | Country of Origin | Study type and participants | Collation and summary of results | MMAT Score (* if one criterion is met; ***** if all five criterions are met) |
|---|---|---|---|---|
| Spiegel et al. (2022) [36] | United States of America | Population-based retrospective cohort study of people with EDS birthing between the years 1999 and 2014 (n = 1042) | • Prevalence of EDS in pregnancy was 7 per 100,000 births, with the trend increasing over the 16-year study period (p < .0001)<br>• Those with EDS were more likely to be Caucasian, belong to a higher income quartile, and smoke.<br>• Pregnancies in those with EDS were associated with prematurity, 1.47 aOR (1.18–1.82), cervical incompetence, 3.11 aOR (1.99–4.85), antepartum hemorrhage, 1.71 aOR (1.16–2.50), placenta previa, 2.26 aOR (1.35–3.77) and maternal death, 9.04 aOR (1.27–64.27)<br>• Those with EDS were more likely to give birth via caesarean section, 1.55 aOR (1.36–1.76), have longer postpartum stays (>7 days), 2.82 aOR (2.08–3.85), and have a neonate with intra-uterine growth restriction, 1.81 aOR (1.29–2.54) | ***** |
| Pezaro et al. (2020) [7] | United Kingdom, United States of America, and Canada | Qualitative interview study including those who had previously given birth and had hEDS/HSD or equivalent diagnosis under a preceding nosology (n = 40). | • Worsening of symptoms reported during pregnancy and postnatal complications.<br>• Anesthesia was often reportedly ineffective.<br>• Long latent phases of labour quickly developed into rapidly progressing active labours and births.<br>• Perinatal staff were observed to be panicked by unexpected outcomes and were deemed to lack the knowledge and understanding.<br>• Poor care resulted in disengagement from services, trauma, stress, anxiety, and an avoidance of future childbearing. | ***** |
| Knoepp et al. (2013) [37] | United States of America | Participants in a longitudinal cohort study of pelvic floor disorders after childbirth (n = 587). | • Hypermobility was diagnosed in 46 (7.8%) and was associated with decreased odds of cesarean birth after complete cervical dilatation or operative vaginal delivery, 0.51 aOR (0.27–0.95).<br>• Anal sphincter laceration was unlikely to occur in those with hypermobility 0.19 aOR (0.04–0.80). | **** |
| Solak et al. (2009) [38] | Turkey | Questionnaire study including pregnant cisgender women (n = 70) and age-matched non pregnant cisgender women (n = 40). | • The prevalence of temporomandibular disorders (TMD) and systemic joint hypermobility in pregnancy were not high compared to age matched non-pregnant people.<br>• 35% with TMD present with general joint hypermobility.<br>• No correlation between systemic joint hypermobility and TMD. | *** |

(*Continued*)

**Table 3.** (Continued)

| Article | Country of Origin | Study type and participants | Collation and summary of results | MMAT Score (* if one criterion is met; ***** if all five criterions are met) |
|---------|-------------------|------------------------------|----------------------------------|------------------------------------------------------------------------------|
| Sundelin et al. (2017) [39] | Sweden | Cohort study of 314 singleton births to those with JHS/EDS prior to childbirth compared with 1 247 864 singleton births to those without a diagnosis of JHS/EDS. | • No elevated risk for preterm birth, preterm premature rupture of membranes, caesarean, stillbirth, low Apgar score, babies being either small or large for gestational age.<br>• Those with Ehlers-Danlos syndrome (n = 62), had a higher risk of induction of labor and amniotomy. | *** |
| Sorokin et al, (1994) [40] | United States of America | Cross sectional survey of 68 cisgender women from the Ehlers-Danlos National Foundation (EDNF) | • Stillbirth rate was 3.15% (3/95).<br>• Preterm birth rate was 23.1% (22/95).<br>• Spontaneous abortion rate was 28.9% (40/138).<br>• Caesarean birth rate of 8.4%.<br>• 14.7% had perinatal bleeding problems. | * |
| Hurst et al, (2014)[9] | United States of America | Cross sectional study including 1,769 members of the Ehlers-Danlos National Foundation; 1,225 with a confirmed diagnosis of EDS | • Rates of obstetric outcomes for those who reported at least one pregnancy included term pregnancy in 69.7%<br>• Preterm birth in 25.2%<br>• Spontaneous abortion in 57.2%<br>• Ectopic pregnancy in 5.1%<br>• Infertility was reported by 44.1% of survey respondent.<br>• Normal menstrual cycles were reported by only 32.8%.<br>• Gynecologic pain reported included dyspareunia by 77.0% | * |
| Lind et al, (2002) [41] | The Netherlands | Retrospective questionnaire study among members of the Dutch Ehlers-Danlos Association. 66 affected cisgender women (all types of EDS) and 33 unaffected. | • High rate of pelvic pain and instability (26% vs 7%).<br>• Maternal complications consisted of pelvic pain and instability (26% vs. 7%).<br>• Preterm birth occurred in 21% of those affected compared with 40% of those nonaffected with an affected infant.<br>• Those with EDS experienced postpartum hemorrhage (19% vs. 7%) and complicated perineal wounds (8% vs. 0%) more often than those unaffected.<br>• Floppy infant syndrome was diagnosed in 13% of the affected infants and did not occur in the nonaffected neonates. | * |

*(Continued)*

**Table 3.** (Continued)

| Article | Country of Origin | Study type and participants | Collation and summary of results | MMAT Score (* if one criterion is met; ***** if all five criterions are met) |
|---|---|---|---|---|
| Karthikeyan, et al (2018) [42] | United Kingdom | Cohort study of those childbearing with hEDS/HSD (n = 8). | • Majority experienced pelvic girdle pain in pregnancy.<br>• One patient suffered from severe gastro-intestinal hypomotility symptoms and required percutaneous endoscopic gastrostomy (PEG) feeds.<br>• Symptoms of gastro-esophageal reflux from hiatus hernia (n = 2).<br>• Obstetrically indicated caesarean section births (n = 5).<br>• Declined vaginal birth due to anxiety about recurrent hip joint dislocations and obstetric cholestasis (n = 1).<br>• Preterm vaginal birth at 29/40 weeks gestation (n = 1).<br>• Vaginal birth at 39/40 weeks gestation (n = 1).<br>• Placental abruption at 31/40 weeks gestation (n = 1).<br>• Regional blockade anesthesia for pain relief in labor/caesarean section was found to be effective. | * |
| **Case Studies** | | | | |
| Atalla, et al. (1988) [43] | United Kingdom | Single case study. | • Increasing joint laxity during pregnancy, required prolonged bed rest and traction.<br>• Birth via caesarean as performed early to relieve symptoms. | N/A |
| Cesare et al. (2019) [44] | United States of America | Single case study of 22-year-old with a history of morbid obesity, seizures, Barrett's esophagus, hypermobility being evaluated for EDS, and anaphylaxis to an unknown local anesthetic, scheduled for cesarean birth. | • After rapid-sequence induction of general anesthesia, video laryngoscopy facilitated endotracheal intubation.<br>• Birth and recovery were uneventful. | N/A |
| De Vos et al. (1999) [45] | Belgium | Single case study of a 33-year-old nullipara referred for preconceptual genetic counselling with a history of easy bruising, generalized joint hypermobility and chronic arthralgia and myalgia. The diagnosis of hEDS was confirmed on clinical examination. | • Prophylactic McDonald cerclage placed in situ at 14 weeks' gestation. Premature rupture of membranes occurred at 23 weeks' gestation.<br>• A female infant was born at 26 weeks following the onset of chorioamnionitis and died 3 h after birth.<br>• Electron-microscopic examination showed collagen fiber abnormalities in the fetal skin, indicative of EDS. | N/A |
| Fedoruk et al. (2015) [46] | Canada | Single case study of a 26-yr-old primigravid person diagnosed with hEDS prior to pregnancy. | • Genetic testing during pregnancy revealed a heterozygous variant of unknown significance in the *COL3A1* gene causative for vascular type EDS.<br>• An induced labour was planned with instrument-assisted vaginal birth.<br>• Epidural catheter used for analgesia during labor and birth. Outcomes were excellent. | N/A |
| Garcia-Aguado et al. (1997) [47] | France | Single case study of a primigravid person with hEDS. | • Obstetrical extradural analgesia administered without complication. | N/A |

(*Continued*)

**Table 3.** (*Continued*)

| Article | Country of Origin | Study type and participants | Collation and summary of results | MMAT Score (* if one criterion is met; ***** if all five criterions are met) |
|---|---|---|---|---|
| Golfier et al. (2001) [48] | France | Single case study of two consecutive pregnancies in a 21-year-old pregnant person. | • Progressive worsening of joint instability and pain during both pregnancies. Uneventful prophylactic lower segment caesarean (n = 2) performed because of hip joint (sub)luxations appearing upon any slight effort of abduction.<br>• Wound healing and use of anesthesia (general and epidural) was unremarkable. | N/A |
| Jones et al. (2008) [49] | United Kingdom | Single case study of a primigravid person with hEDS/HSD associated with POTS. | • Single-shot spinal anesthesia was performed after failed epidural anesthesia. | N/A |
| Kanjwal et al. (2009) [50] | United States of America | A single case study of a 37-year-old with JHS who developed symptoms of recurrent syncope in the postpartum period. | • POTS presenting 6-months postpartum. | N/A |
| Morales-Roselló et al. (1997) [51] | Spain | A single case study of pregnancy with hEDS/HSD. | • Uneventful and favorable outcome. | N/A |
| Ogawa et al. (2022) [52] | Japan | A single case study of a 20-year-old primigravid person with a dichorionic diamniotic twin pregnancy in which Ehlers-Danlos syndrome was first suspected at 19 weeks gestation | • Inpatient bed rest from 29 weeks of gestation to manage a worsening of hip pain present since early pregnancy.<br>• Hip pain improved with bed rest, but back and pelvic pain gradually worsened.<br>• Birth occurred via an elective caesarean 34 weeks gestation.<br>• Both infants were healthy at birth (suspected joint hypermobility)<br>• Postpartum depression, referred to the psychiatry department. | N/A |
| Place et al. (2017) [53] | United Kingdom | Single case study of a 24-year-old primiparous person with POTS, systemic lupus erythematous and hEDS/HSD | • Underwent induction of labor at 36+5 weeks pregnant. Uterus became hyperstimulated.<br>• Birth via caesarean clinically indicated.<br>• Risk of intrathecal blockade mitigated by: (i) using the lateral position when placing the spinal anesthetic, (ii) giving a fluid bolus prior to placing the spinal to provide adequate preload and (iii) titrating phenylephrine both to heart rate and systolic blood pressure. | N/A |
| Quak et al. (2013) [54] | United Kingdom | Single case study of A 19-year-old nulliparous person pregnant with a tongue base T-lymphoblastic lymphoma, JHS and a history of LA insensitivity. | • Pregnancy uneventful other than persistent symptoms of stridor.<br>• Uneventful birth via caesarean for anesthetic indications. | N/A |
| Roop et al.(1999) [55] | United States of America | Case report of two infants affected by hEDS. | • Abnormal presentation noted during pregnancy and reduced birth weight observed. | N/A |
| Sakala et al. (1991) [56] | United States of America | Single case report of two pregnancies experienced by the same patient with hEDS/HSD. | • Both pregnancies ended in uneventful vaginal births at 37>+ weeks gestation. Outcomes were good. | N/A |
| Selcer et al. (2021) [57] | United States of America | A single case study of a 22-year-old pregnant with hEDS/HSD and a long history of chronic pain, recurrent joint dislocations in several joints, signs of autonomic dysfunction, irritable bowel, history of dysmenorrhea and heavy menses, osteopenia, and family history of flexibility. | • On fetal ultrasound, an enlarged fetal aorta was noted.<br>• Fetal echocardiogram at 27 weeks and 2 days revealed mild dilation of the aortic root and there was also dilation of the ascending aorta. | N/A |

(*Continued*)

**Table 3.** (Continued)

| Article | Country of Origin | Study type and participants | Collation and summary of results | MMAT Score (* if one criterion is met; ***** if all five criterions are met) |
|---------|-------------------|----------------------------|----------------------------------|------------------------------------------------------------------------------|
| Sizer, (2014) [58] | United States of America | A single case study of a 21-year-old with hEDS/HSD who developed progressive decline in mobility, and physical conditioning during her pregnancy with her first child. | • Chronic pain and recurrent subluxations were modestly responsive to prolotherapy.<br>• Pain and hypermobility were managed with opioid pain medications, therapy, and bracing.<br>• Frequency of hip subluxation increased in the second trimester of pregnancy.<br>• Changes in body shape and centre of gravity precluded consistent wear of her orthotics.<br>• Mobility was achieved with a manual wheelchair. Intensive aqua therapy with transition to land-based therapy arrested the decline in conditioning and free movement until the birth of the baby. | N/A |
| Sood et al. (2009) [59] | United Kingdom | Single case study of general anesthesia being used in a pregnant patient with hEDS. | • General anesthesia with rapid sequence induction was performed for caesarean section due to prolonged second stage of labor. However, intubation proved to be difficult.<br>• Patient resistant to local anesthesia. | N/A |
| Taylor et al. (1981) [60] | England | Single case report of an uneventful pregnancy in a patient with hEDS. | • Uneventful birth occurred via caesarean due to fixed hips. | N/A |
| Khalil et al. (2013) [61] | Australia | Single case study of a 34-year-old primigravida with hEDS | • Referred to a geneticist at 20 weeks of pregnancy (confirmed hEDS/HSD)<br>• Received care from both the cardiac and obstetric teams throughout pregnancy.<br>• Cardiology review at 28 weeks of pregnancy was organized and 12-lead echocardiogram (ECG) and aortic root was within normal limits.<br>• Antenatal care uneventful and labor was induced at 40 weeks after spontaneous rupture of membranes.<br>• A healthy baby was delivered by emergency caesarean, performed due to slow progress in labour.<br>• Spinal analgesia used for the caesarean.<br>• No complications an unremarkable recovery.<br>• Entonox, pethidine, and intravenous paracetamol used for pain relief after the obstetric anesthetist consultant advised against the use of epidural analgesia. | N/A |
| Volkov et al (2006) [62] | Israel | Single case study of a 32-year-old diagnosed with hEDS | • An uneventful vaginal birth at term. | N/A |
| Leduc et al. (1992) [63] | United States of America | Single case study: cervical incompetence due to defective connective tissue was treated with a Smith-Hodge pessary | • At 33 weeks' gestation the cervix had dilated to 5 cm, the membranes had ruptured, and contractions began.<br>• 470 gm male born vaginally over a small midline episiotomy.<br>• The total time in labor was 4 hours.<br>• Blood loss was 250 cc.<br>• Episiotomy scar healed well.<br>• Infant had Apgar scores of 7 and 9 and 1 and 5 minutes and was normal. | N/A |

**Table 4. Expert guidelines.**

| Overview | Through pregnancy, birth and beyond, multi-modal, comprehensive, and integrative treatment strategies are required to ensure every approach to the management of hEDS/HSD remains individualized. It is important to listen to those personally affected and involve them in decision making at every level. Management options include, but are not limited to, medications, interventional procedures, physical therapy, nutrition, mental health support, and complementary therapies.<br><br>Multidisciplinary teams are required and may usefully include specialist input as appropriate from a variety of qualified professionals who are educated in and knowledgeable of considerations with EDS e.g.<br>Obstetricians<br>Midwives<br>Physicians<br>Physician's associates/assistants<br>Nurse practitioners<br>Rehabilitation practitioners (physical therapists / occupational therapists/ chiropractors)<br>Anesthesiologists<br>Rheumatologists<br>Immunologists<br>Geneticists<br>Endocrinologists<br>Pain specialists<br>Clinical nutritionists or dietitians<br>Mental health teams<br>Sleep specialists (including neurologists, pulmonologists)<br>Clinical researchers |
|---|---|

| Preconceptual Guidelines | | | |
|---|---|---|---|
| **Category** | **Caveats** | **Action Points** | **Supporting literature** |
| Genetic Counselling | • HEDS/HSD is characterized by genetic heterogeneity; no molecular genetic testing is currently available.<br>• A parent with hEDS/HSD has an increased chance of having a child with hEDS/HSD. Predicting symptoms is not possible due to intra-familial heterogeneity. | | [3,64–67] |
| Pain Management | Pain is a recognized feature amongst all EDS types and HSD and should be considered before, during and after pregnancy using a comprehensive integrative approach for best effects to improve quality of life and mental wellbeing. Reducing pain interference with healthcare guided and self-care activities is important to prevent deconditioning and an exacerbation of fatigue.<br><br>Pain control is an important part of an EDS patients' quality of life. Stopping and/or reducing pain medications as well as sleep medications and interventions in pregnancy can have significant consequences–worsening overall symptoms and potentially causing be long-term consequences. | • A thorough risk/benefit assessment should be undertaken on an individual basis.<br>• Manual therapy in conjunction with co-interventions (e.g., functional training) may provide short-term improvements in pain and disability in pelvic girdle pain.<br>• Consider evaluation from pelvic pain and/or rehabilitation specialist as appropriate.<br>• Complementary and alternative medicine should be considered as part of the multimodal approach to managing pain (e.g., acupuncture), though the usual contraindications associated with pregnancy still apply. | [7,41,58,68,69] |

**Table 4.** (Continued)

| Comorbidities | • There are multiple comorbidities, often occurring in common with hEDS/HSD, that may influence assessment and treatment (e.g., migraine). These include postural orthostatic hypotension syndrome (POTS), clonal and non-clonal mast cell activation syndrome (MCAS), irritable bowel syndrome (IBS) and endocrine dysregulation. | Screen for symptoms of these conditions: <br>• Clonal and non-clonal mast cell activation syndrome (MCAS). <br>• Irritable Bowel syndrome (IBS); See www.theromefoundation.org/rome-iv/rome-iv-criteria/ <br>• As pregnancy and birth exerts a higher demand on bodily systems, serum screening related to the hypothalamus/pituitary access especially morning cortisol, and adrenocorticotropic hormone (ACTH) may be considered. If screening during pregnancy, Free Cortisol (not Total) should be considered to avoid being impacted by the rising Estrogen effect on cortisol binding globulin. <br>• Orthostatic screening for dysautonomia; See http://www.dysautonomiainternational.org and https://www.nhs.uk/conditions/postural-tachycardia-syndrome/ and http://www.dysautonomiainternational.org/pdf/RoweOIsummary.pdf | [8,10–13,32,38,70–77] |
|---|---|---|---|
| Sexual Intercourse | Anticipate: <br>• Cases of subluxation and/or dislocation during sexual intercourse. <br>• Pain, swelling, or microtears with sexual intercourse | • Prior to pregnancy, in order to participate in sexual intercourse, some individuals may require treatment such as: lubricants, lidocaine 2%, clindamycin and/or estrogen creams internally, Cromolyn, H1 blockers vaginally delivered and/or compounded topical or internal muscle relaxers. Improving proprioception to the best possible status before pregnancy. <br>• Once pregnant, consult with the clinical team as some medications may be contraindicated. <br>• Intimacy (positioning, support of major joints to ensure comfort with use of pillows, wedges and towel rolls, breathing techniques, muscle relaxation techniques) Alternative positioning options. Use of stabilization/strengthening exercises where appropriate. Accommodate for increased estrogen levels which will further relax muscles and mechanical weight distributions which will change during pregnancy. | [9,35,78] |
| Conception | As always, the choice as to whether one should start a family is individual and personal. | | |

| **Prenatal Guidelines** | | | |
|---|---|---|---|
| **Category** | **Caveats** | **Action Points** | **Supporting literature** |
| Risk Assessment | Management plans should be made on a case-by-case basis following a comprehensive and individualized risk assessment. Although some symptoms of pregnancy may be exacerbated, it is important to understand that pregnancy and birth may be unremarkable. | • Discuss awareness of possible complications and outcomes relating to hEDS/HSD in initial screening questionnaires to ensure appropriate consultation and monitoring early on | [7,9–13,33–37,40–43,61,63] |
| Miscarriage and termination of pregnancy | The overall risk of miscarriage in those with hEDS/HSD is not significantly greater than in the general population. There is however an increased risk for recurrent miscarriage (≥3 with the same partner) in a subgroup with hEDS/HSD. <br><br>Anticipate increased and prolonged bleeding, often due to platelet function abnormalities in this population | • Consider using desmopressin and/or tranexamic acid if excessive bleeding. | [7,9,34–36,39–41,79,80] |

*(Continued)*

**Table 4.** (Continued)

| Prenatal management | There is conflicting evidence regarding the risk of preterm labor, preterm birth, and cervical insufficiency in those with hEDS/HSD.<br><br>Anticipate:<br>• Increased bleeding during pregnancy due to increased tissue fragility and platelet dysfunction.<br>• Increased risk of fetal growth restriction.<br><br>There is no known increased risk of:<br>• Uterine rupture/torsion.<br>• Pre-eclampsia.<br>• Stillbirth.<br>• Clinically significant mitral valve prolapse or aortic root dilation in those pregnant with hEDS/HSD.<br><br>There is no evidence to support:<br>• The use of prophylactic cerclage.<br>• Routine use of progesterone in the 2nd /3rd trimester.<br>• Bedrest. | • A cervical length screen should be performed at the time of an anatomic ultrasound in the second trimester.<br>• Use desmopressin and tranexamic acid if excessive bleeding.<br>• A single echocardiogram may be considered for screening purposes; echocardiography is not needed if the aortic root has been identified as normal prior to pregnancy.<br>• Those with known aortic dilation should have an echocardiogram in each trimester; a cardiologist should be involved their care.<br>• Anesthesia for delivery should be preplanned, particularly where there is temporomandibular joint (TMJ) dysfunction, spinal issues, dysautonomia, and/or mast cell activation (see www.tmsforacure.org). | [7,10,33–36,41,42,45,52,55,61,63,79,81–83] |
|---|---|---|---|
| Mobility | Symptoms related to connective tissues may be significantly affected by the hormones supporting pregnancy. This may affect one's mobility with hEDS/HSD during pregnancy in a variety of ways.<br><br>**Anticipate increased:**<br>• Joint laxity.<br>• Pelvic Girdle Pain (PGP), particularly in early pregnancy.<br>• Low back pain.<br>• Hip pain.<br>• Pelvic and joint instability.<br>• Muscle spasms. | • Referral to physical therapy and other rehabilitation specialists.<br>• Bracing, support, and assistive devices as indicated for functional activity and gait during pregnancy.<br>• Reduction of pain and limitations with activities in daily living through: individualized stability exercises, education and manual therapy techniques to address pain, instability/hypermobility, weakness, muscle spasm and restrictions in muscle, fascia and blood flow.<br>• Modified core strengthening exercises.<br>• Due to connective tissue laxity with EDS, joint mobilization / manipulation may be harmful and only should be used with caution. Manual therapy interventions should utilize various direct and indirect techniques (including but not limited to: soft tissue mobility, myofascial release, positional release/indirect myofascial techniques).<br>• Avoid applying manual therapies in isolation.<br>• *See additional resources section, see supplement: physical therapy*. | [5–7,10,41,42,52,58,84–95] |

(*Continued*)

**Table 4.** (Continued)

| | | | |
|---|---|---|---|
| Gastrointestinal Issues | **Anticipate the following:**<br>• Increase in small intestinal bacterial overgrowth (SIBO) due to changes in gut bacteria and increase in constipation during pregnancy.<br>• Increased malabsorption and digestion issues influenced by gastroparesis/nutrient co-factor/ change in medications/ chewing/reflux.<br>• Increased intestinal methane overgrowth (IMO)<br><br>**Worsening of:**<br>• Gastroparesis/delayed gastric emptying.<br>• Sluggish bowel motility gastritis.<br>• Irritable Bowel syndrome (IBS).<br>• Diarrhea (due to hormonal pregnancy changes and alterations in frequency/serving size of food intake). | • If significant weight loss, consider oral supplementation, then enteral or parenteral supplementation as last resort.<br>• Constipation, managed as per usual guidelines (e.g., increased soluble fiber/fluids, 2 kiwi fruit per day, probiotic supplementation and physical therapy as required). Fiber-rich foods can be pureed or chopped and better tolerated to help with constipation.<br>• Nausea, managed as per usual guidelines (e.g., small frequent meals, ginger, peppermint, electrolyte fluid). Monitor zinc status if symptoms persist. Lower fiber, monitor weight.<br> • If significant weight loss, consider oral supplementation, then enteral or parenteral supplementation as last resort.<br>• Gastro-esophageal reflux and dyspeptic symptoms<br> • Sodium alginate capsules for GERD/LPR treatment<br> • Eat small, frequent meals, avoid foods that exacerbate GERD and any other GI symptoms if indicated.<br> • Elevate the head of bed with risers underneath feet of bed, not with a wedge pillow. | [5–7,11–13,42,96,97] |
| Postural Orthostatic Hypotension Syndrome (POTS)/Dysautonomia | **Anticipate the following:**<br>• Increased symptoms of POTS in early pregnancy if hyperemesis and dehydration present.<br>• Improved symptoms of POTS as hemodilution occurs later in pregnancy, offsetting some of the symptoms associated with low blood volume.<br>• Anticipate increased heart rate and other exacerbated symptoms of dysautonomia during sexual intercourse.<br>• Some individuals may experience an initial onset of POTS during pregnancy. | • Exercises tailored to improving one's ability to remain upright.<br>• Use of compression hosiery at 30mmHg–40mmHg.<br>• Consider α1-receptor agonist (midodrine-though animal studies document decreased fetal size and increased fetal demise) or beta blockers.<br>• Consider stress dose steroids if on fludrocortisone medications.<br>• Maintain adequate hydration, at least 2.5 L (10.6 cups of fluid daily) e.g., water, low sugar electrolyte replacement drinks, broth.<br>• Salt/sodium intake should be tailored to the individual's medical history, conditions, medication interactions.<br>• Salt (5-7g)/sodium (2000–2800 mg) general goal.<br>A range of nutritional deficiencies are commonly seen among those with POTS, especially those who are malnourished and/or have significant gastrointestinal diseases. In cases of antenatal malnutrition, guidelines include:<br>• Testing for deficiencies in vitamin A, D, E, K, iron, folate, B12, thiamine and essential fatty acids.<br>• Dietary adjustments or supplements based on test results and symptoms.<br>• If a deficiency of vitamin A is identified, a strategy to safely replete through food and/or supplements per the WHO guidelines is required. | [10,32,50,73,74,98] |

(*Continued*)

**Table 4.** (Continued)

| Mast Cell Activation Syndrome (MCAS) | **Anticipate the following:**<br>• Worsening of MCAS symptoms such as rhinitis, chronic spontaneous urticaria, asthma, and neuropsychiatric conditions as mast cells are triggered by physical/psychological stress.<br>• Increased mast cell reactions caused (for example) by breast/chest feeding or sexual intercourse (e.g., itching (pruritus), hives (urticaria), swelling (angioedema), skin flushing, wheezing, shortness of breath and/or stridor with throat swelling.<br>• Dysregulation in number/function of mast cells due to hormonal changes.<br>• 'Brain fog' with MCAS.<br>• Potential chronic spontaneous urticaria (CSU), pruritic urticarial papules (hives) and plaques in pregnancy (PUPPP). | • Specialized medical evaluation and assistance to reduce risk of mast cell activation events, due to natural, as well as medications or procedures.<br>• Action plans to prevent mast cell activation events (e.g., pre-medication and pre-hydration protocols).<br>• Individualized plans to recognize and treat acute symptoms, such as anaphylaxis and asthma exacerbations.<br>• Prolonged and active medical observation post childbirth<br>• Screening recommendations:<br>  • Complete Blood Count (CBC) with differential measures<br>  • Baseline immunoglobulin (IG) levels<br>  • Spirometry test for asthma | [75–77] |
|---|---|---|---|
| **Intrapartum Guidelines** | | | |
| **Category** | **Caveats** | **Action Points** | **Supporting literature** |
| Risk assessment | Births in hEDS/HSD populations are physiologically straightforward in the majority of cases.<br>However, anticipate the following:<br>• Fetal malpresentation<br>• Unstable fetal lie<br>• Protracted latent phase of labor/prodromal or 'false' labor.<br>• Rapid active stage/precipitous vaginal birth (≤3 hours) with a frequency of 28–36%<br>• Increased bleeding due to increased tissue fragility and platelet dysfunction.<br>• Increased risk of urinary retention | • Consider using desmopressin and/or tranexamic acid if excessive bleeding.<br>• Consider temporary catheterization of bladder if urinary retention occurs. | [7,9–13,33–37,40–43,45,46,51–57,62,63,79,83,99,100] |
| Birth choices: Mode of birth | In all contexts, aim to create calming environments to optimize physiological processes and reduce adverse outcomes, pain, and anxiety. Gentle handling of fragile tissues is important in all cases. Shared decision making between clinical team and birth parent is essential in all cases.<br><br>Physiological vaginal birth is clinically preferable where there are no obstetric contraindications.<br>• Anticipate and prepare for a protracted latent phase of labor/prodromal or 'false' labor followed by precipitous (≤3hrs) active labor with an increased risk of perineal trauma (due to increased tissue fragility).<br>• Assisted vaginal birth is not contraindicated.<br><br>Birth via cesarean should be reserved for where there are obstetric indications only. However, consider cesarean when:<br>• Extensive vaginal tearing is anticipated and there is a history of tissue fragility to reduce the risk of fissures and possibly pelvic prolapse.<br>• Joint issues precluding positioning for vaginal birth, e.g., fixed, painful. | For vaginal birth:<br>• Use manual perineal protection (MPP) unless intervention declined or incompatible with positioning at birth.<br>• Use spontaneous rather than directed pushing techniques.<br>• Use instruments with caution due to increased tissue fragility and considering that the neonate may also display symptoms of hEDS/HSD such as bruising and hypotonia.<br><br>For caesarean birth:<br>• Slow-dissolving sutures and glue (abdominal wall) are recommended. | [7,10,33,34,46,48,51,53–56,60,62,73] |

*(Continued)*

**Table 4.** (Continued)

| Birth Choices: Intended place of birth | In all cases where one's intended place of birth choices are being considered, it is important to understand local contexts, service provision and individual circumstances. | In accordance with patient preferences and to manage precipitous births occurring prior to patients reaching their intended place of birth more effectively, it may be prudent to plan for a home birth, either as a contingency or where appropriate. | [7,10,33,34,46,51,52,54–56,62] |
|---|---|---|---|
| Mobility and positioning in labor | Optimal positioning when giving birth can promote optimal outcomes. Those with hEDS/HSD can be challenged in their mobility due to joint laxity and difficulty with weight distribution preventing and inhibiting movement.<br>• Anticipate joint articulation pain or dysfunction problems; this may necessitate additional input from physical therapy. | • Carefully position head and neck, shoulders, back, hips, knees, and ankles due to increased risk of pain and dislocations as well as spinal instabilities.<br>• Use positioning aids (e.g., wedges, exercise balls, pillows, bed adaptations) to assist where necessary.<br>• Ensure clinical team is receptive and listening to the individual birthing regarding how they feel that they are able to and want to move their own bodies at any given time. | [7,10,39,41,42,52,59,73,101–103] |
| Anesthesia | Those childbearing with hEDS/HSD may require anesthesia whether for general pain relief or surgery. Anesthetic plans should be made in advance of childbirth.<br>**Anticipate the following:**<br>• Temporomandibular joint dysfunction and cervical spine instability may make intubation and airway management more difficult.<br>• Subluxation and/or dislocation of any joint<br>• Higher doses of local anesthesia may be needed.<br>• Possibility of cerebrospinal fluid (CSF) leak after spinal anesthesia<br>• POTS:<br> • Consider early analgesia and close monitoring of hemodynamic status.<br> • Epidural is preferred over spinal anesthesia (less likely to cause sudden changes in systemic vascular resistance and subsequent instable hemodynamics). Phenylephrine may reduce the risk of reactive tachycardia. | • Increase mobilization and consider use of a sequential compression device (SCD)/ compression hosiery.<br>• If general anaesthesia is necessary, be aware of possible co-morbid conditions such as cranio-cervical instabilities and Chiari I malformations.<br> • The head should be positioned during intubation in a careful manner remaining aware of anatomical positions, including in emergencies.<br>• To mitigate a or cerebral spinal fluid (CSF) leak, consider earlier/prophylactic administration of blood patch due to increased tissue fragility and complex healing in this population. If headaches are present in the postnatal period consider CFS leak, whether or not a lumbar puncture has been performed. | [7,10,38,42,44,46–49,53,54,59,61,73,74,100,104–107] |

| Postnatal Guidelines | | | |
|---|---|---|---|
| **Category** | **Caveats** | **Action Points** | **Supporting literature** |
| Wound Healing | Broadly, those with hEDS/HSD typically experience poor wound healing and complex tissue scarring (e.g., atrophic scar or keloid formation). As there is often a need for healthy wound and tissue healing following childbirth, it is important to consider the management of healing distinctly in these populations.<br><br>**Anticipate the following:**<br>• Delayed healing of all wounds (e.g., perineal, caesarean, and other surgical) wounds<br>• Abnormal scar formation (e.g., following episiotomy or caesarean)<br>• Pelvic prolapse.<br>• Suture failure | • Use longer lasting, absorbable sutures. Use additional adhesives in cases where skin is particularly fragile.<br>• Close all wounds without tension, using interrupted suturing.<br>• Referral to physical therapy specialists for management of scar. | [7,10,34,37,62,108] |

(*Continued*)

**Table 4.** (Continued)

| | | | |
|---|---|---|---|
| Pelvic health | Those with hEDS/HSD in recovery from pregnancy and childbirth can experience delays in regaining their pelvic health (pelvic floor, abdominal, hip, and back regions) and increased levels of pelvic pain as the connective tissues behave differently in these populations.<br><br>**Anticipate the following:**<br>• Pelvic venous disorders associated with chronic pelvic pain.<br>• Dislocation and/or subluxations of any pelvic joint including but not limited to the coccyx; sacroiliac joint; and symphysis pubis<br>• Increased rates of pelvic organ prolapse and fissures.<br>• Prolonged/chronic Pelvic Girdle Pain (PGP)<br>• Pelvic floor dysfunction | • Evaluate pelvic venous disorder symptoms, e.g., venous compression.<br>• Pelvic health professional skilled with EDS where symptoms are identified.<br>• Rehabilitation assessment for dislocation by a qualified professional<br>• First line treatment includes manual techniques to address muscle and fascia soft tissue involvement.<br>• In cases of chronic/prolonged PGP use repeated intra-pelvic corticosteroid injection treatments with caution due to tissue/bone degradation<br>• Joint fusion may be necessary in some cases.<br>• Appropriate referrals regarding the following:<br>  • Urinary incontinence, urgency, frequency, and dysfunction<br>  • Pelvic organ prolapse.<br>  • Constipation, bowel incontinence<br>  • Dyspareunia<br>  • Pain (pelvic, pelvic girdle pain, pubic symphysis) | [7,9,10,34,37,41,95,109–113] |
| Care of the newborn | When caring for the newborn of those with hEDS/HSD it is important to consider that they may also experience symptoms and later diagnoses related to hypermobility. Diagnoses should be made in line with the paediatric diagnostic framework for paediatric joint hypermobility.<br><br>**Anticipate increases in the following:**<br>• Bleeding<br>• Hematoma<br>• Joint subluxation, particularly following manipulation at birth including but not limited to:<br>  • Clavicle<br>  • Hip<br>  • Neck | • Documentation of early bruising and causes to avoid false accusations of maltreatment.<br>• Offer positional support to birth parent and infant in cases of joint instability.<br>• Consult rehabilitation professionals to assist with modifications in body mechanics, posture, and environmental set up, splinting, bracing or assistive devices as appropriate. | [7,10,41,52,114–116] |

*(Continued)*

Table 4. (Continued)

| | | | |
|---|---|---|---|
| Infant feeding | Where human milk feeding is preferred, additional teaching, logistical and physical support may be required for those with hEDS/HSD to avoid placing undue strain on joints or other connective tissues affected. Breast/Chest feeding can counteract diuresis following childbirth, particularly useful in those with POTS.<br><br>**When breast/chest feeding anticipate the following:**<br>• Fatigue and pain can impact one's ability to maintain optimal feeding positions.<br>• Increased pain, head, and neck strain where positions remain unsupported.<br>• MCD: Increased mast cell reactions can be caused by breast feeding/chest feeding.<br>• Monitor pain and safety of medication(s) for breastfeeding/chest feeding.<br>• Consider safety when the infant also shows signs of hypermobility and/or low tone. | • Identify suitable positions when breast/chest feeding to avoid sudden movements and possible injury.<br>• Utilization of props, bracing, assistive devices, and pillows as needed.<br>• Mixed feeding (artificial/human milk) where appropriate.<br>• Avoid positions which hyperextend joints in arms/hands.<br>• Experiment with positions laying down (avoid falling asleep).<br>• Monitor and treat cracked nipples.<br>• Breast/chest feed or bottle feed infant in a comfortable chair with good back support that allow feet to be comfortable on the ground.<br>• Keep the baby well supported at breast/chest height.<br>• Latch position: Check with mirror rather than straining head and neck.<br>• Continuously evaluate optimal positioning while supporting back, neck, head, shoulders, pelvis, and lower extremities. | [7,73,74,117] |
| Psychological wellbeing and mental health | Psychological wellbeing and mental health is relevant throughout the perinatal period and beyond. People with hEDS/HSD have historically been classified as "somatizers", and can perceive hostility and disinterest from clinicians, further exacerbating psychological distress in this population. Any condition featuring chronic pain may be accompanied by anxiety, depression, and suicidal ideation. Existing mental health conditions may also be exacerbated by pregnancy. While associations between hEDS/HSD attention-deficit/hyperactivity disorder (ADHD) and autism spectrum disorder (ASD) have been observed, more robust evidence in this area is required to explore these in more depth.<br><br>**Anticipate the following:**<br>• Decreased health-related quality of life.<br>• Increased psychological distress and symptoms of anxiety and depression.<br>• Delay in help-seeking and/or hypervigilance.<br>• Higher rates of postpartum psychosis and post-traumatic stress disorder (PTSD). | • Comprehensive mental health risk assessment, screening and planning adopting a multidisciplinary approach to providing individualised care.<br>• Compassionate communication validating hEDS/HSD experiences.<br>• Shared decision making.<br>• Referral to local and/or national mental health services as appropriate. | [10,118–122] |

(*Continued*)

**Table 4.** (Continued)

| Additional resources | <ul><li>www.hEDSTogether.com</li><li>Principles of Physical Therapy—Dr. Leslie Russek</li><li>Statement on the use of opioids in pain management of the Ehlers-Danlos syndromes and hypermobility spectrum disorders</li><li>Broad guidelines on the management of pain during the antenatal and postpartum period</li><li>Broad guidelines for the administration of analgesia and anesthesia whilst breastfeeding/chest feeding</li><li>Genetic and Rare Diseases (GARD) Information Center Website</li><li>Symptoms and triggers of Mast Cell Activation</li><li>Perioperative Management—TMS—The Mast Cell Disease Society, Inc: TMS–The Mast Cell Disease Society, Inc (tmsforacure.org)</li><li>TMS_ER-Protocol-2022.pdf (tmsforacure.org)</li><li>www.theromefoundation.org/rome-iv/rome-iv-criteria/</li><li>http://www.dysautonomiainternational.org/pdf/RoweOIsummary.pdf</li><li>http://www.dysautonomiainternational.org</li><li>Clinical Practice Guidelines for Pelvic Girdle Pain in the Postpartum Population</li><li>ACOG Committee Statement: Physical Activity and Exercise During Pregnancy and the Postpartum Period</li><li>Pelvic Girdle Pain in the Antepartum Population: Physical Therapy Clinical Practice Guidelines Linked to the International Classification of Functioning, Disability, and Health from the Section on Women's Health and the Orthopaedic Section of the American Physical Therapy Association</li><li>World Health Organization 2020 guidelines on physical activity and sedentary behaviour</li><li>Pelvic Pain.org</li><li>https://www.iahp.com/pages/search/index.php</li></ul> | | |

pregnancy and mobility), Intrapartum (risk assessment, birth choices, mode of birth/intended place of birth, mobility in labor and anesthesia), and Postpartum (wound healing, pelvic health, care of the newborn and infant feeding). Our use of the wider literature in co-creation adds to earlier syntheses and understandings in this area [124,125]. Yet further research is required with regards to obstetric outcomes dependent upon the pregnant person's and the unborn' EDS status, particularly where outcomes in relation to specific types of EDS are not reported separately [82,126].

Due to their inextricable links, PPI activities and co-creators also directed us to provide expert guidelines in relation to some of the comorbidities of hEDS/HSD such as mast cell diseases and spinal instability [127]. Equally, these guidelines evolved to encompass dysautonomia, orthostatic intolerance with or without orthostatic hypotension and POTS, of which between 70% to ≥80% of those with hEDS/HSD may also experience symptoms related to childbearing [127]. Guidelines were also included in relation to pain management, mental health, nutrition, supplementation, allergies with immune responses, and overall wellbeing. Further resources and citations are also provided and should be referred to, particularly in relation to paediatrics [116].

In the assessment of risk in planning both the mode and place of birth, the anticipation of increased blood loss will be key in all cases. Individuals with hEDS/HSD typically have fragile capillaries and tissue, predisposing them to bruising and hematomas. In addition, some will have an abnormal interaction between Von Willebrand factor, platelets, and collagen resulting in suboptimal blood clotting which in turn can lead to heavier and prolonged bleeding [128]. During pregnancy, increased mucosal fragility can result in spontaneous bleeding (e.g., epistaxis and gingival). There may similarly be increased bleeding during childbirth [10,34,40,41]. Yet some large retrospective studies have found no increased risk for bleeding in such cases [35,36]. Nevertheless, alongside other more well-known treatments, we have been able to endorse a wider variety of medicines such as Desmopressin and Tranexamic Acid to control bleeding where applicable [99,100]. Due to the potential concern for precipitous birth in this population [7,10], it may be prudent to plan for births in a community setting with attention to distance to any birthing facilities.

It is useful to consider that the extensibility of all bodily tissues in those childbearing with hEDS/HSD may be greater than in the general childbearing population. During pregnancy, the entire body must be considered since pregnancy hormones such as relaxin influence the body systemically [128]. Those with hEDS/HSD can also present with a lower health related quality of life and greater psychological distress than those in the general population [118], which may be exacerbated by childbearing [10]. Some patients experience a relief in symptoms, particularly dysautonomia and pain. Due to the multifaceted nature of hEDS/HSD, a biopsychosocial approach may be most appropriate in all cases, whereby symptoms such as depression and anxiety are always assessed concurrently with physical symptoms and treated accordingly. In this task, a suite of co-created tools to help perinatal staff support people childbearing with hEDS/HSD may usefully be employed [14].

As connective tissues behave differently for those birthing with hEDS/HSD, it may be useful to investigate outcomes relating to alternate forms of analgesia (e.g., waterbirth) in these populations. Moreover, it is possible that some features of childbearing considered 'typical' may actually be associated with hEDS/HSD (e.g., precipitous birth), corresponding with joint hypermobility, skin hyperextensibility and other anomalies of connective tissue [10]. Future research activities could explore these potential links and avoid conflation between the various subtypes of EDS.

Using systematic and co-creation approaches along with a variety of PPI activities, this research is the first of its kind to offer consensus guidelines from an international and

multidisciplinary group of experts in the field of childbearing with hEDS/HSD. Limitations include a lack of relevant, larger, longitudinal, and high-quality studies in this field. Those with hEDS/HSD may not have been diagnosed until after their childbearing experience, thus limiting clinical expertise in this area. Further research is required to compare outcomes of interventions designed to address all conditions more prevalent in this childbearing population (e.g., PGP) [128].

Due to conflicting findings and limited research in this area, the guidelines presented have also been guided by patient preference, clinical expertise, practicalities and known biological mechanisms. Inevitably, as evidence-based practitioners, some co-creators have also referenced other published evidence, for example in cases where we have made evidence-based guidelines for the use of complementary and alternative medicines such as acupuncture [69], though the usual contraindications associated with pregnancy still apply [129]. Furthermore, many research findings and guidelines published previously have been made for all EDS types, rather than for individuals with hEDS/HSD specifically. Thus, we have had to review and modify these in light of the wider evidence and expertise within the co-creation team. The quality of some evidence such as clinical case studies was low. Moreover, our scoping review is inherently limited as its focus was to provide breadth rather than depth of information. Many existing guidelines for pregnancy in general populations (e.g., in relation to exercise and pelvic girdle pain) remained relevant and considered equally beneficial with modification(39). Yet in other areas such as cervical insufficiency and preterm labor, further research will be required for more comprehensive guidelines.

A key challenge in conducting this research is that the inheritance of hEDS/HSD has yet to be determined, unlike the other types of EDS. Also, multiple family members may have hEDS/HSD, with significant variability in findings and symptoms among them. Whole exome sequencing has recently identified several genes of interest [65], but at this time there is no genetic testing available. Genetic factors may also play a role where incidence of multiple miscarriages of pregnancy ($\geq$3 with the same partner) in this population is higher(30). Limitations in the literature include the fact that diagnoses of hEDS/HSD are often self-reported, and studies frequently report on all subtypes of EDS as a whole, where some of the rarer types (e.g., vascular) would likely bring increased complications. As hEDS/HSD does represent the vast majority of cases, it is likely representative of the complications reported. Yet even a minority of complications included from one of the rarer types of EDS in an amalgamated cohort study would skews results and thus, the interpretation of risk. Future research could usefully embark upon prospective studies which explore each subtype distinctly. Moreover, despite the risks and issues discussed in the guidelines presented here, it remains important not to over medicalize pregnancy, as many pregnancies in cases of hEDS/HSD are decidedly unremarkable.

## Conclusion

The co-creation of evidence-based guidelines in the management of childbearing in cases of hEDS/HSD is justified given the lack of knowledge and awareness demonstrated by healthcare professionals and the profound impacts hEDS/HSD can have upon childbearing populations. Evidence in this area was found to be sparse and somewhat contradictory. Nevertheless, the expert guidelines established here by clinical, academic, and patient experts provide an evidence-informed basis for how care and outcomes may be improved for this childbearing population. In the spirit of evidence-based practice, such guidance could usefully evolve alongside future research in this field.

## Other information

SP reports receiving honorariums from the Ehlers-Danlos Society. AH reports receiving honorariums from the Ehlers-Danlos Society. NB reports receiving honorariums from the Ehlers-Danlos Society. Other authors report no conflict of interest. All data extracted has been included in this report.

## Supporting information

**S1 File. PRISMA-ScR.** PRISMA extension checklist for Scoping Reviews.
(PDF)

**S2 File. Supporting information.** Guidelines for physical therapy.
(DOCX)

## Acknowledgments

The authors wish to thank Dr Maarten (M.F.M.) Engel from the Erasmus MC Medical Library for their support in developing and updating the search strategies. We would also like to thank the experts we consulted throughout, and the Ehlers Danlos Society.

## Author Contributions

**Conceptualization:** Sally Pezaro, Isabelle Brock, Natalie Blagowidow.

**Data curation:** Sally Pezaro, Isabelle Brock, Maggie Buckley, Sarahann Callaway, Serwet Demirdas, Alan Hakim, Cheryl Harris, Carole High Gross, Megan Karanfil, Isabelle Le Ray, Laura McGillis, Bonnie Nasar, Melissa Russo, Lorna Ryan, Natalie Blagowidow.

**Formal analysis:** Sally Pezaro, Isabelle Brock, Maggie Buckley, Sarahann Callaway, Serwet Demirdas, Alan Hakim, Cheryl Harris, Carole High Gross, Megan Karanfil, Isabelle Le Ray, Laura McGillis, Bonnie Nasar, Melissa Russo, Lorna Ryan, Natalie Blagowidow.

**Investigation:** Sally Pezaro, Isabelle Brock, Maggie Buckley, Sarahann Callaway, Serwet Demirdas, Cheryl Harris, Carole High Gross, Megan Karanfil, Isabelle Le Ray, Laura McGillis, Bonnie Nasar, Melissa Russo, Lorna Ryan, Natalie Blagowidow.

**Methodology:** Sally Pezaro, Isabelle Le Ray.

**Project administration:** Sally Pezaro.

**Validation:** Isabelle Le Ray, Melissa Russo.

**Writing – original draft:** Sally Pezaro, Alan Hakim, Natalie Blagowidow.

**Writing – review & editing:** Sally Pezaro, Isabelle Brock, Maggie Buckley, Sarahann Callaway, Serwet Demirdas, Alan Hakim, Cheryl Harris, Carole High Gross, Megan Karanfil, Isabelle Le Ray, Laura McGillis, Bonnie Nasar, Melissa Russo, Lorna Ryan, Natalie Blagowidow.

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
