## [Decision Letter · Decision Letter 0]

24 Mar 2024

PONE-D-23-42863Management of Childbearing with Hypermobile Ehlers-Danlos Syndrome and Hypermobility Spectrum Disorders: A Scoping Review and Expert Co-creation of Evidence-based GuidelinesPLOS ONE

Dear Dr. Pezaro,

Thank you for submitting your manuscript to PLOS ONE. After careful consideration, we feel that it has merit but does not fully meet PLOS ONE’s publication criteria as it currently stands. Therefore, we invite you to submit a revised version of the manuscript that addresses the points raised during the review process.

Your manuscript was well-received by reviewers.  Please address by revision of the manuscript or rebuttal of each of the points below.  

We look forward to receiving your revised manuscript.

Kind regards,

Martin E. Matsumura, MD

Academic Editor

PLOS ONE

[SP reports receiving honorariums from the Ehlers-Danlos Society. AH reports receiving honorariums from the Ehlers-Danlos Society. NB reports receiving honorariums from the Ehlers-Danlos Society. Other authors report no conflict of interest.]. 

Reviewers' comments:

Reviewer's Responses to Questions

**Comments to the Author**

1. Is the manuscript technically sound, and do the data support the conclusions?

Reviewer #1: Yes

Reviewer #2: Yes

2. Has the statistical analysis been performed appropriately and rigorously? 

Reviewer #1: Yes

Reviewer #2: I Don't Know

3. Have the authors made all data underlying the findings in their manuscript fully available?

Reviewer #1: Yes

Reviewer #2: Yes

4. Is the manuscript presented in an intelligible fashion and written in standard English?

Reviewer #1: Yes

Reviewer #2: Yes

5. Review Comments to the Author

Reviewer #1: Excellent summary and recommendations, which are greatly needed. The hypermobility population, including patients and physicians, will appreciate your hard work on this publication. You may consider recommendations for methylated folate instead of folic acid supplementation, but this is not necessary for publication.

This publication is going to help many!

Reviewer #2: The authors underwent a process to create management guidelines for those with hEDS/HSD related to childbearing from conception, pregnancy, delivery, and post-partum concerns. They relied on literature review that was comprehensive involving several databases and languages; however, the literature was limited. The approach also involved experts in the various fields as well as patients themselves to create these guidelines. While the literature has pointed out some recommendations, these guidelines tried to be complete in looking at pregnancy in its entirety but with important perspectives from experienced professionals and patients themselves. The approach was vigorous trying to use acceptable standards in such guideline development as well as fills a much needed clinical gap.

Unfortunately, the limitations of the literature also include EDS often reported by the patients themselves (which may or may not be correct) as well as including EDS as a whole which would include all types of which classic and vascular would likely have increased complications. As HSD/hEDS does represent the vast majority of EDS types, it is likely representative of the complications reported but a few cases of complications from one of the rarer types of EDS would skew results and interpretation of risks. It is difficult to get around this without prospective studies. This is a major limitation of this guideline and should be discussed more thoroughly.

It is hoped that the mention of these many issues does not over medicalize pregnancy when as the authors mentions, most pregnancies go as expected. I would suggest that this is also addressed as many patients also have misconceptions about pregnancy and so do health providers. I have heard many tell me that their doctors have told them to never get pregnant, an abhorrent misconception of the risks for hEDS/HSD.

Recommendation is to accept with revisions:

Major:

The guidelines recommend consideration of midodrine use during pregnancy for those with dysautonomia. The US FDA lists this as pregnancy category C with animal studies documenting decrease fetal size and increased fetal demise. For the expected physiologic response to this drug as a vasoconstrictor, this is very plausible. Clinical experience is limited reporting no detrimental outcomes but this is not unusual [Morgan et al., “POTS and Pregnancy”, Int J Women’s Health, 2022]. Recommendation without expressing this concern is potentially harmful especially as this guideline is to inform clinical decision-making.

Many of the references do not list a journal title which is not appropriate.

Minor:

As the guidelines use a compilation of articles related to symptomatic joint hypermobility which includes hEDS and HSD as well as joint hypermobility syndrome. Statement addressing JHS as unable to be distinguished from hEDS/HSD is necessary as this guideline should also apply to those diagnostically labeled as JHS which is considered likely allelic to hEDS and definitely within the hypermobility spectrum disorders.

Table 1 reports data from a published source. The title reports “Signs and symptoms of EDS…” therefore it is not clear if this relates to all type of EDS or hEDS/HSD specifically. This citation could not be verified using the information provided. The data should also outline if this is for females only or a preponderance of females as expected for this population.

Table 3, Castori et al. 2012 reference: in the 4th column and 4th bullet point, it should read “births at >37 weeks gestation (66.7%)”

Table 3, Holick et al. 2021 reference: unfortunately, this was focused on the birth outcome when the fetus/infant had another genetic condition altogether. It should be deleted as irrelevant as not to give the impression that fetal fractures are a rare outcome of hEDS.

Table 3, Sizer et al. 2014 reference: 3rd column lists mobility as decreased. As the manuscript talks about joint hypermobility this is a confusing term suggesting to the less avid reader that the joints can acquire decrease mobility during pregnancy (when it is often increased) when this is talking about ambulation of the person. This is also used confusingly in the manuscript itself and alternate terminology would be useful.

Table 4, Overview: I was surprised to see geneticist not listed as part of the multidisciplinary team especially as the discipline was included in the co-creators. Of course, not every patient needs all specialists but many geneticists may help manage these patients. A geneticist may also be helpful in discussing the diagnosis and heritability risks as well as discussion on testing as appropriate since many may be referred for ruling out the rarer forms of EDS.

Table 4, Comorbidities: adrenal insufficiency is listed which in my experience of thousands of patients is not the case. Diagnostic studies are often negative but patients do show abnormal cortisol levels which may be due to many other concerns. If the authors contend that this is a co-morbidity and can worsen during pregnancy, should not an endocrinologist be listed as a team member?

Table 4, Sexual Intercourse: phrasing “cromolyn and/or compounded topical or internal muscle relaxers”. Cromolyn is not a muscle relaxer. I suspect that the sentence/phrasing is incomplete.

Table 4, Miscarriage: column 2 has “due to due to” is repetitive.

Table 4, wound healing: column 2, “suture dehiscence” is unclear. Is it meant suture failure or wound dehiscence. Both would be appropriate.

Table 4, Additional resources: website “Statement on the use of opioids…” used twice

6. PLOS authors have the option to publish the peer review history of their article (what does this mean?). If published, this will include your full peer review and any attached files.

Reviewer #1: **Yes: **Jacques Courseault, MD

Reviewer #2: No

---

## [Author Response · Author response to Decision Letter 0]

28 Mar 2024

Response to reviewers

Dear reviewers,

Many thanks for your constructive feedback. Your requests for changes will no doubt improve the quality of our manuscript further prior to publication. We have addressed them point by point in the table below, and highlighted changes in yellow within the revised manuscript.

Reviewer Comment Our response

Reviewer #1: Excellent summary and recommendations, which are greatly needed. The hypermobility population, including patients and physicians, will appreciate your hard work on this publication. You may consider recommendations for methylated folate instead of folic acid supplementation, but this is not necessary for publication.

This publication is going to help many! Thank you.

No consensus exists over supplementing methylated folate over folic acid in HSD/hEDS. However, in cases where there exists clinical concern regarding methylenetetrahydrofolate reductase activity due to a MTHFR polymorphism and where a low serum folate has been identified, supplementation with biologically active 5-MTHF (also known as L-5-MTHF, 5-methyl-folate, L-methylfolate, and methylfolate) may be considered1. Supplementation with (5-MTHF) may be specifically preferable in cases with of megaloblastic risk2. Outside these parameters, prenatal methylfolate as preference to folic acid is not advised3. To avoid overcomplication and as folic acid is broadly recommended in all routine pregnancy care, we have not added this to the guidelines presented here.

1. Carboni L. Active Folate Versus Folic Acid: The Role of 5-MTHF (Methylfolate) in Human Health. Integr Med (Encinitas). 2022 Jul;21(3):36-41. PMID: 35999905; PMCID: PMC9380836.

2. Ferrazzi E, Tiso G, Di Martino D. Folic acid versus 5- methyl tetrahydrofolate supplementation in pregnancy. Eur J Obstet Gynecol Reprod Biol. 2020 Oct;253:312-319. doi: 10.1016/j.ejogrb.2020.06.012. Epub 2020 Jun 13. PMID: 32868164.

3. https://uktis.org/monographs/use-of-methylfolate-in-pregnancy/

Reviewer #2: The authors underwent a process to create management guidelines for those with hEDS/HSD related to childbearing from conception, pregnancy, delivery, and post-partum concerns. They relied on literature review that was comprehensive involving several databases and languages; however, the literature was limited. The approach also involved experts in the various fields as well as patients themselves to create these guidelines. While the literature has pointed out some recommendations, these guidelines tried to be complete in looking at pregnancy in its entirety but with important perspectives from experienced professionals and patients themselves. The approach was vigorous trying to use acceptable standards in such guideline development as well as fills a much-needed clinical gap. Thank you.

We have been able to cite 2 additional pieces of literature which we have since become aware of to support our discussions further. However, these have not resulted in any changes to our guidelines or scoping review findings.

Tofts LJ, Simmonds J, Schwartz SB, Richheimer RM, O’Connor C, Elias E, Engelbert R, Cleary K, Tinkle BT, Kline AD, Hakim AJ. Pediatric joint hypermobility: a diagnostic framework and narrative review. Orphanet journal of rare diseases. 2023 May 4;18(1):104.

This paper has led to a slight revision to the newborn section of the guideline where we have directed the reader to use these diagnostic criteria. “. Diagnoses should be made in line with the paediatric diagnostic framework for paediatric joint hypermobility.”

Alrifai, N., Alhuneafat, L., Jabri, A., Khalid, M. U., Tieliwaerdi, X., Sukhon, F., ... & Sharma, T. (2023). Pregnancy and fetal outcomes in patients with Ehlers-Danlos syndrome: a nationally representative analysis. Current Problems in Cardiology, 48(7), 101634.

Unfortunately, the limitations of the literature also include EDS often reported by the patients themselves (which may or may not be correct) as well as including EDS as a whole which would include all types of which classic and vascular would likely have increased complications. As HSD/hEDS does represent the vast majority of EDS types, it is likely representative of the complications reported but a few cases of complications from one of the rarer types of EDS would skew results and interpretation of risks. It is difficult to get around this without prospective studies. This is a major limitation of this guideline and should be discussed more thoroughly. Indeed – we do allude to this earlier in the discussion section but have now added this detail more thoroughly in the limitations discussed at the end of the discussion.

Limitations in the literature include the fact that diagnoses of hEDS/HSD are often self-reported, and studies frequently report on all subtypes of EDS as a whole, where some of the rarer types (e.g., vascular) would likely bring increased complications. As hEDS/HSD does represent the vast majority of cases, it is likely representative of the complications reported. Yet even a minority of complications included from one of the rarer types of EDS in an amalgamated cohort study would skews results and thus, the interpretation of risk. Future research could usefully embark upon prospective studies which explore each subtype distinctly. 

It is hoped that the mention of these many issues does not over medicalize pregnancy when as the authors mentions, most pregnancies go as expected. I would suggest that this is also addressed as many patients also have misconceptions about pregnancy and so do health providers. I have heard many tell me that their doctors have told them to never get pregnant, an abhorrent misconception of the risks for hEDS/HSD. 

Agreed – we have reiterated this at the end of the discussion section.

Moreover, despite the risks and issues discussed in the guidelines presented here, it remains important not to over medicalize pregnancy, as many pregnancies in cases of hEDS/HSD are decidedly unremarkable.

The guidelines recommend consideration of midodrine use during pregnancy for those with dysautonomia. The US FDA lists this as pregnancy category C with animal studies documenting decrease fetal size and increased fetal demise. For the expected physiologic response to this drug as a vasoconstrictor, this is very plausible. Clinical experience is limited reporting no detrimental outcomes but this is not unusual [Morgan et al., “POTS and Pregnancy”, Int J Women’s Health, 2022]. Recommendation without expressing this concern is potentially harmful especially as this guideline is to inform clinical decision-making. Thank you – important point.

We have added this directly into the guideline section as “(midodrine- though animal studies document decreased fetal size and increased fetal demise)”

As you rightly point out – these are clinical guidelines and so we have added ‘clinical’ to the article title to reflect this

Many of the references do not list a journal title which is not appropriate. We have used the Endnote reference style for Plos One – Journal titles have been added where available.

As the guidelines use a compilation of articles related to symptomatic joint hypermobility which includes hEDS and HSD as well as joint hypermobility syndrome. Statement addressing JHS as unable to be distinguished from hEDS/HSD is necessary as this guideline should also apply to those diagnostically labeled as JHS which is considered likely allelic to hEDS and definitely within the hypermobility spectrum disorders. Indeed – Thank you.

We have added this in to the first paragraph of the discussion section.

“as JHS is indistinguishable from hEDS/HSD and considered likely allelic, this guideline should also apply to those diagnostically labelled as having JHS.”

Table 1 reports data from a published source. The title reports “Signs and symptoms of EDS…” therefore it is not clear if this relates to all type of EDS or hEDS/HSD specifically. This citation could not be verified using the information provided. The data should also outline if this is for females only or a preponderance of females as expected for this population. Thank you

We have changed the title of this table to add clarity 

“Signs and symptoms of EDS (all subtypes) in females with a frequency ≥70% (34 out of 79 signs)”

Table 3, Castori et al. 2012 reference: in the 4th column and 4th bullet point, it should read “births at >37 weeks gestation (66.7%)” Corrected in the table – thank you

Table 3, Holick et al. 2021 reference: unfortunately, this was focused on the birth outcome when the fetus/infant had another genetic condition altogether. It should be deleted as irrelevant as not to give the impression that fetal fractures are a rare outcome of hEDS. Thank you – this reference has now been removed through the guideline and manuscript. This has inevitably altered number sequencing throughout.

Table 3, Sizer et al. 2014 reference: 3rd column lists mobility as decreased. As the manuscript talks about joint hypermobility this is a confusing term suggesting to the less avid reader that the joints can acquire decrease mobility during pregnancy (when it is often increased) when this is talking about ambulation of the person. This is also used confusingly in the manuscript itself and alternate terminology would be useful. Thank you

For clarity we have reframed this as “free movement”

Table 4, Overview: I was surprised to see geneticist not listed as part of the multidisciplinary team especially as the discipline was included in the co-creators. Of course, not every patient needs all specialists but many geneticists may help manage these patients. A geneticist may also be helpful in discussing the diagnosis and heritability risks as well as discussion on testing as appropriate since many may be referred for ruling out the rarer forms of EDS. We have now included geneticists and endocrinologists in line with this and the below comment – Thank you.

Table 4, Comorbidities: adrenal insufficiency is listed which in my experience of thousands of patients is not the case. Diagnostic studies are often negative but patients do show abnormal cortisol levels which may be due to many other concerns. If the authors contend that this is a co-morbidity and can worsen during pregnancy, should not an endocrinologist be listed as a team member? Thank you – we have consulted with an endocrinologist throughout and have rephrased this section again in light of your comment.

It is now reframed as Endocrine dysregulation. The main citation we use to justify this is:

Casanova EL, Sharp JL, Edelson SM, Kelly DP, Sokhadze EM, Casanova MFJb. Immune, autonomic, and endocrine dysregulation in autism and Ehlers-Danlos syndrome/hypermobility spectrum disorders versus unaffected controls. 2019:670661.

Table 4, Sexual Intercourse: phrasing “cromolyn and/or compounded topical or internal muscle relaxers”. Cromolyn is not a muscle relaxer. I suspect that the sentence/phrasing is incomplete. We have now rephrased this sentence to avoid confusion.

Table 4, Miscarriage: column 2 has “due to due to” is repetitive. Thank you – repetition deleted.

Table 4, wound healing: column 2, “suture dehiscence” is unclear. Is it meant suture failure or wound dehiscence. Both would be appropriate. Thank you – we have changed this now to read ‘suture failure’

Table 4, Additional resources: website “Statement on the use of opioids…” used twice Thank you – repetition removed.

---

## [Editor Report · Decision Letter 1]

3 Apr 2024

Management of childbearing with hypermobile Ehlers-Danlos Syndrome and Hypermobility Spectrum Disorders: A scoping review and expert co-creation of evidence-based clinical guidelines

PONE-D-23-42863R1

Dear Dr. Pezaro,

We’re pleased to inform you that your manuscript has been judged scientifically suitable for publication and will be formally accepted for publication once it meets all outstanding technical requirements.

Kind regards,

Martin E. Matsumura, MD

Academic Editor

PLOS ONE

Additional Editor Comments (optional):

Thank you for a thorough revision of your work based on reviewer comments!